# SPOP in Cancer: Phenomena, Mechanisms and Its Role in Therapeutic Implications

**DOI:** 10.3390/genes13112051

**Published:** 2022-11-07

**Authors:** Xiaojuan Yang, Qing Zhu

**Affiliations:** Department of Abdominal Oncology, West China Hospital of Sichuan University, Chengdu 610041, China

**Keywords:** speckle-type POZ protein, modular structures, tumorigenesis, biological processes, treatment

## Abstract

Speckle-type POZ (pox virus and zinc finger protein) protein (SPOP) is a cullin 3-based E3 ubiquitin ligase adaptor protein that plays a crucial role in ubiquitin-mediated protein degradation. Recently, *SPOP* has attracted major research attention as it is frequently mutated in a range of cancers, highlighting pleiotropic tumorigenic effects and associations with treatment resistance. Structurally, SPOP contains a functionally critical N-terminal meprin and TRAF homology (MATH) domain for many SPOP substrates. SPOP has two other domains, including the internal Bric-a-brac-Tramtrack/Broad (BTB) domain, which is linked with SPOP dimerization and binding to cullin3, and a C-terminal nuclear localization sequence (NLS). The dysregulation of SPOP-mediated proteolysis is associated with the development and progression of different cancers since abnormalities in SPOP function dysregulate cellular signaling pathways by targeting oncoproteins or tumor suppressors in a tumor-specific manner. SPOP is also involved in genome stability through its role in the DNA damage response and DNA replication. More recently, studies have shown that the expression of SPOP can be modulated in various ways. In this review, we summarize the current understanding of SPOP’s functions in cancer and discuss how to design a rational therapeutic target.

## 1. Introduction

Protein degradation is an essential biological process that is involved in cellular homeostasis. Many diseases, including various cancers, result from the abnormal accumulation of proteins [1]. The ubiquitin proteasome system controls approximately 80% of the ubiquitination and degradation of intracellular proteins [2,3,4]. Ubiquitination is a conserved post-translational modification that is involved in regulating cell proliferation, differentiation, cell cycle progression, apoptosis, transcription, DNA damage and repair and drug resistance [2,3,5]. Ubiquitination is closely associated with carcinogenesis resulting from oncogenic alterations that disrupt the ubiquitination of proteins involved in key tumor suppressor pathways [6]. Proteasome-mediated degradation is a multi-step process that involves the labelling of target proteins with ubiquitin molecules (a single ubiquitin protein or multiple ubiquitin molecules), followed by the degradation of the target ubiquitinated substrates by the 26S proteasome complex [7,8].

Briefly, the first step in proteasomal degradation is the ATP-dependent activation of ubiquitin by the ubiquitin-activating E1 enzyme. The activated ubiquitin is transiently conjugated to a ubiquitin-conjugating E2 enzyme through a transthiolation reaction, after which a ubiquitin-protein E3 ligase transfers to the specific target substrate for ubiquitin ligation [9]. In mammalian cells, different numbers of ubiquitin ligases exist, including several E1 ligases, 30–40 E2 ligases and more than 600 putative E3 ubiquitin ligases [10,11]. Among these E3, the Cullin–RING ubiquitin ligases (CRLs) complex family is the most prominent group and consists of eight Cullin scaffold proteins (CRL1, 2, 3, 4A, 4B, 5, 7, 9) (CRL1, 2, 3, 4A, 4B, 5, 7, 9) [10,12,13]. Generally, the Cullin–RING ligase complex consists of the Cullin scaffold protein, a ring box protein, an adaptor, and the substrate receptor. In contrast to other CRLs, the Cullin3-RING consists of three primary components: the Cullin scaffold protein, a ring box protein called RBX1 and a Bric-a-brac-Tramtrack/Broad (BTB) protein that performs the dual functions of an adaptor and substrate receptor [13,14]. The speckle-type POZ protein (SPOP) is produced by the *SPOP* gene and was originally identified as a protein by Nagai et al. in 1997. SPOP is a BTB protein containing meprin and TRAF homology (MATH) and BTB domains that can bind Cullin 3 and other substrates [15]. 

An increasing amount of evidence indicates that SPOP may have dual functions in tumorigenesis. *SPOP* variants have been reported in many types of cancer, including prostate (PrCa) [16,17,18], endometrial cancer (EC) [19], ovarian [20,21], liver [22], thyroid [23,24], breast [25] and kidney cancers [26] amongst others. The majority of SPOP-related studies have been carried out in PrCa, where it has been reported that 6–15% of patients have point mutations in the substrate-binding MATH domain despite the discovery of a mutation in the BTB domain [27,28]. Mutations in the MATH domain have been recognized as an early event in the development and progression of PrCa, as *SPOP* mutations are associated with PrCa’s lack of rearrangements of TMPRSS2 ETS family genes [29,30,31]. 

Several genomic analyses have reported *SPOP* mutations in EC [32,33,34]. A previous study showed that *SPOP* mutations in EC fail to interact with and ubiquitinate estrogen receptor-α (ERα), thus facilitating tumorigenesis [35]. In contrast, a recent study suggested that *SPOP* mutations associated with EC increase the ability of three BETs (BRD2, BRD3 and BRD4) to bind and enhance polyubiquitination of these substrate proteins, leading to degradation and enhanced sensitivity to BET inhibitors [19]. These studies highlight that the pathophysiological function of *SPOP* mutations remains to be fully understood in EC. 

*SPOP* mutants associated with ovarian, liver and thyroid cancers have been reported in several sequencing studies [20,22,23,24]. In breast cancer, studies have revealed that SPOP targets substrates such as c-Myc, steroid receptor coactivator 3 (SRC-3) and progesterone receptor (PR), thus functioning as a tumor suppressor [25,36,37]. However, SPOP can also directly interact with and mediate the ubiquitination and degradation of breast cancer metastasis suppressor 1 (BRMS1), which is a member of the metastasis suppressor family with known repressive effects on distant metastasis [38]. In contrast to other cancer types, studies have shown that SPOP is highly expressed in clear-cell renal cell carcinomas (RCCs). SPOP-mediated tumorigenesis in RCCs may be mediated by the degradation of cell proliferation suppressors and anti-apoptosis regulators such as Phosphatase and tensin homolog (PTEN), Daxx, Dual-Specificity Phosphatase 7 (DUSP7) and Gli2 [39]. These findings suggest that SPOP can act as a tumor suppressor in various solid cancers but also as a tumor promoter in RCCs. The molecular mechanism through which SPOP performs its functions is that its substrates mediate various cellular processes, including signaling pathways, transcriptional regulation, genome stability and so on. Additionally, SPOP can be regulated by MicroRNAs (miRNAs) and other molecules and can be post-transcriptionally modified by cationic processes such as phosphorylation and self-ubiquitination. 

In this review, we introduce the structural features of SPOP and provide a foundation for understanding that SPOP interacts with and mediates efficient substrate ubiquitination in multiple cancers. We discuss the substrate-regulated processes that are involved and the underlying molecular mechanisms of oncogenesis. Finally, we discuss the regulatory mechanisms of *SPOP* gene expression.

## 2. The Modular Structure of SPOP

### 2.1. The SPOP MATH Domain

SPOP is a 374-residue protein that consists of four domains: an N-terminal MATH domain, a BTB/POZ domain, a BACK domain and a C-terminal NLS (Figure 1). The MATH domain (residues 31–164) has an antiparallel β-sandwich structure that selectively recruits substrates and is the predominant domain in which mutations occur [40]. Several critical residues such as Y87, F102, Y123, W131 and F133 are capable of binding the SPOP-binding consensus (SBC) motif from different substrates through the shallow central cleft [14,40]. The SBC motif comprises a five-residue Φ-π-S-S/T-S/T (Φ, denotes a nonpolar amino acid; π, denotes a polar amino acid) [40,41,42]. The interactions of the MATH domain and SBC motif rely on hydrophobic and polar interactions [40]. It has been reported that the binding affinity of one domain is independent of the other MATH domain within the dimer. Consequently, the dimeric configuration can increase the ability of binding to substrates in a substrate-specific manner, but only these substrates contain multiple suboptimal SBC motifs [6,43]. Additionally, it is important that polyubiquitination be followed by degradation when substrates have multiple SBC motifs, since substrates merely undergo monoubiquitination when they contain only one SBC motif [43]. Zhuang et al. reported that SBC phosphorylation in MacroH2A and Puc can block binding to SPOP, however; more studies are needed to verify these observations [40]. Exploratory analyses of SBC phosphorylation of other substrates at perhaps a range of sites and the responding biochemical functions are required.

### 2.2. The SPOP BTB Domain 

The BTB domain is a versatile protein-protein interaction motif and a common structural element that was discovered in zinc finger transcription factors and Cul3 substrate adaptors. In the human genome, it is encoded by ~205 genes [8,44]. The BTB domain (residues 184–297) is responsible for SPOP dimerization and interactions. Specifically, an α3–β4 loop that consists of approximately ten amino acid residues in the BTB domain and is fundamental for the interactions [10,45,46,47]. 

Beyond the BTB domain, α-helices make up the three-box domain, which can enhance SPOP-Cul-3 interactions [48]. The BTB domain can dimerize to promote the dimerization of SPOP-cullin3 to generate an oligomeric CRL3 that is highly multivalent and has multiple catalytic centers for substrates [40,48,49]. There is a very low dissociation constant within BTB domain-mediated dimerization [43]. The dimerization capability is associated with α1-3 helices and is related to a strand-exchanged amino terminal region. Four essential hydrophobic BTB dimerization residues, L186, L190, L193 and I217, jointly contribute to the residues 184–297 to form SPOP dimers [14,40]. Dimerization-defective *SPOP* mutants exhibit impaired ubiquitination without a significant decrease in binding cullin3 affinity [40]. In the human proteome, only SPOP and its homolog, SPOP-like (SPOPL), have both BTB and MATH domains, despite BTB usually being linked to other interaction domains [44]. 

### 2.3. The SPOP BACK Domain

The BACK domain (residues 300–359) also mediates dimerization [49]. There are over 50 known human proteins that have BACK and BTB domain combinations [44]. Only the BACK domains within SPOP and SPOPL (which is a unique human SPOP analog) have an atypical truncation that allows dimerization of the BACK domain [50]. The dimerization interface consists of the BTB and BACK domains. The C-terminus acts independently to form higher-order SPOP oligomers capable of enhancing the ubiquitination of substrates [51]. These oligomers enable augmenting the E3 activity by increasing the substrate avidity and the availability of the E2 ubiquitin-conjugating enzyme [48,51]. 

### 2.4. The SPOP NLS Domain 

SPOP contains a NLS at its C-terminus, in which amino acids 359–374 are essential for its location at nuclear speckles. The NLS is essential for the nuclear localization of SPOP and its interactions with nuclear-localizing substrates. SPOP lacking the NLS accumulates in the puncta of the cytosol [52]. 

## 3. SOP Expression in the Development, Progression and Treatment of Cancer 

Multiple studies have suggested that SPOP has dual functions in tumorigenesis based on gene expression and clinical datasets. The analysis of datasets from The Cancer Genome Atlas (TCGA) and other sources has indicated that the majority of cancers have significantly decreased expression of SPOP compared to corresponding normal controls, including bladder urothelial carcinoma (BLCA), colon adenocarcinoma (COAD), kidney chromophobe (KICH), lung adenocarcinoma (LUAD), lung squamous cell carcinoma (LUSC), prostate adenocarcinoma (PRAD), rectum adenocarcinoma (READ), thyroid carcinoma (THCA) and uterine corpus endometrial carcinoma (UCEC) [Figure 2a]. 

Cholangiocarcinoma (CHOL), esophageal carcinoma (ESCA), liver hepatocellular carcinoma (LIHC), kidney renal clear cell carcinoma (KIRC) and kidney renal papillary cell carcinoma (KIRP) show a marked increase in the expression of *SPOP*, yet the ESCA variable was not statistically significant [Figure 2a]. 

Analysis of data in the cBioPortal database (www.cbioportal.org/) has shown that *SPOP* alterations occur in different tumor types. The alterations include mutation, structural variants, amplification, deep deletion and multiple alterations [Figure 2b]. *SPOP* mutations are most common in PrCa, in which the *SPOP* mutation rate is 12%, which is consistent with previous reports [27]. A previous clinical study suggested that *SPOP* mutations and downregulation were tightly associated with a poor prognosis in patients with PrCa [18]. From the data presented in Figure 2a, further analyses revealed that the upregulation of SPOP may predict favorable outcomes in KIRC and LUAD. In contrast, the low expression of SPOP is associated with improved outcomes in KICH, KIRP, LIHC and PRAD [Figure 2c]. However, patients with BLCA, CHOL, COAD, LUSC, READ, THCA and UCEC do not have significant associations between SPOP expression and overall survival (OS) [Figure 2c]. These findings require further validation before being used in routine patient management. In addition to the observations made in clinical cancer datasets, previous studies have also found that SPOP can have both oncogenic and tumor-suppressor functions [41].

In 2010, Kan et al. originally reported that *SPOP* is frequently mutated in PrCa [53]. Subsequently, whole-genome and exome sequencing analyses discovered that *SPOP* mutations occur as early events in the development of PrCa but do not occur in normal issues or prostate stroma [17,27,29,30,54,55]. In PrCa, *SPOP* mutations are located in the MATH domain, which typically contains Y87, F102, S199, F125, W131, K129, F133 and K143 mutations [Figure 1] that have been identified as biologically relevant events [27]. *SPOP* mutations can lead to genomic instability and impaired genome maintenance, resulting in the inactivation of BRCA1 and an impaired homology-directed repair (HDR) function [55]. Importantly, several have further indicated that SPOP acts as a tumor suppressor by promoting the ubiquitination and subsequent degradation of downstream substrates, including the Androgen receptor (AR) [56,57], steroid receptor coactivator 3 (SRC3) [58], TRIM24 [59], Gli3 [60,61,62,63], HIPK2 [64], 53BP1 [65], (Programmed death ligand 1) (PD-L1) [66], BMI1 [67], Macrohistone H2A1 (MacroH2A) [67,68], Pancreatic duodenal homeobox 1 (Pdx1) [69,70,71], SENP7 [72], ERG [73,74], BRD2/3/4 [19,75,76], DEK [77], DDIT3 [78], Nanog [79,80], Cdc20 [81], CYCLIN E1 [82], c-MYC [83], INF2 [84], EgIN2 [85], Activating Transcription Factor 2 (ATF2) [86], fatty acid synthase (FASN) [87], Caprin1 [52], 17βHSD4 [88], ITCH [89], GLP [90], CDCA5 [91], PDK1 [92], SQSTM1 [93] and PrLZ [94]. These substrates can regulate diverse cellular processes in PrCa [Figure 3a]. colorectal cancer, diffuse large B-cell lymphoma, lung cancer, liver cancer, choriocarcinoma, Ewing sarcoma and hepatoblastomthe downregulation of *SPOP* can promote the development and progression of PrCa, making it a potential therapeutic target.

The majority of cancer studies involving SPOP have been in PrCa; however, multiple studies have reported *SPOP* mutations in other cancer types including EC, breast cancer (BC), gastric cancer (GC), colorectal cancer, diffuse large B-cell lymphoma, lung cancer, liver cancer, choriocarcinoma, Ewing sarcoma and hepatoblastoma. The SPOP substrates in these cancers include: ERα [35,95], and BRD 2/3/4 in endometrial cancer [19], ERα [96], PR [37], SRC3 [25], c-MYC [36], BRMS1 [38], ASCT2 in breast cancer [97], Gli2 in gastric cancer [98], Gli2 [99], HDAC6 in colorectal cancer (CRC) [100], and MyD88 and CHAF1A in diffuse large B-cell lymphoma (DLBLC) [101,102,103,104]. Fas-associated death domain protein (FADD) [105], SIRT2 in lung cancer (LC) [106], SENP7 in hepatocellular carcinoma (HCC) [107], DHX9 in choriocarcinoma [108], FLI1 in Ewing sarcoma [109], and SLC7A1 [110], DRAK1 and PIPKIIβ in cervical cancer (CC) [111,112]. Figure 3b summarizes the different SPOP substrates and their functions in cellular processes reported in different cancers. The in-depth investigations are necessary to validate the definitive role of SPOP in these cancer types before designing a clear rationale for therapeutic strategies. 

SPOP could potentially have tumor-suppressive roles in several cancers. Emerging evidence demonstrates an oncogenic function of SPOP in kidney cancer (KC). Studies have reported the overexpression and accumulation of SPOP in renal cell carcinoma (RCC) in primary and metastatic tissues that is significantly associated with EMT and poor prognosis [26,113,114,115]. In contrast, *SPOP* silencing in the A498 and ACHN RCC cells induces cell apoptosis, inhibits cell viability, and decreases colony formation and migration ability whilst also increasing sensitivity to Sorafenib [116]. In addition, Li et al. reported that the SPOP protein can accumulate in the cytoplasm under hypoxia to promote tumorigenesis via the degradation of tumor suppressors such as PTEN, DUSP7, ERK phosphatases and Gli2 as well as the proapoptotic protein Daxx in clear cell renal cell carcinoma (ccRCC) [39]. SPOP can also be responsible for the polyubiquitination of the tumor suppressor SETD2 and can negatively regulate H3K36me3 levels in RCC [117,118]. In summary, SPOP possesses an oncogenic role in RCC via overexpression, cytoplasmic accumulation and subsequent ubiquitination and degradation of multiple tumor suppressive substrates [Figure 3c]. Given these findings, SPOP inhibitors may be a potential new therapeutic in the treatment of RCC. 

SPOP has oncogenic or tumor suppressive functions that are cancer type specific. The development of targeted therapies enables treatment for different human cancers. Previous structural analyses have indicated that SPOP binds to the SBC motif of specific substrates via its N-terminal MATH domain [40]. On this basis, Guo et al. identified 109 small molecule inhibitors by computational screening that can disrupt SPOP-substrate interactions [119]. Among the identified molecules, compound 6a was considered an initial hit for binding SPOP that competed with the puc_SBC1 peptide. Additional synthetic optimization of 6a resulted in a compound that had strong inhibitory activity in disrupting SPOP’s binding to PTEN and DUSP7. This resulted in the inactivation of downstream signaling pathways such as the PTEN-phosphoinositide 3-kinase/AKT pathway and the dephosphorylation of extracellular signal-regulated kinase (ERK) [119]. The small-molecule inhibitor 6b was also cytotoxic in A498 ccRCC cells and stabilizes SPOP substrates [119]. Recently, Rincon et al. identified novel therapeutic natural compounds that can specifically target stable MCF-7 SPOP knockdown cells by disrupting GLI3-dependent SHH signaling [120].

## 4. SPOP-Regulated Processes

SPOP serves as a regulatory hub that mainly works on two necessary processes, specifically intracellular signaling pathways and the regulation of genomic function (Table 1) [14]. 

### 4.1. SPOP-Related Intracellular Signaling Pathways

Increasing evidence has shown that SPOP substrates are implicated in the biological processes of cells, including signaling pathways that are essential to maintain normal cellular functions. For example, the Hedgehog (Hh) signaling pathway is a largely conserved signal transduction pathway during metazoan evolution that intimately regulates cell growth and differentiation [63,121]. Cai et al. showed that SPOP plays a critical role in skeletal development and remodeling and that *SPOP* mutations can increase the level of the repressor Gli3R, which has been further explored as a substrate of SPOP but not Gli2 [63]. As a result, Ihh signaling is compromised, suggesting an important positive regulator in SPOP in Hh signaling [63]. These researchers also demonstrated that *SPOP* mutations lead to a significant increase in the level of the activator Gli3, which promotes Shh signaling, suggesting a negative role for SPOP in cord patterning [122]. 

### 4.2. SPOP’s Role in Genome Stability

Impaired DNA damage repair (DDR) can result in genome instability. Initially, it was reported that SPOP participates in DDR in a predominantly ATM-dependent manner [123]. SPOP can be recruited to DNA double-strand break (DSB) foci, where it partially colocalizes with γ-H2AX foci in response to ionizing irradiation (IR) or the topoisomerase 1 inhibitor, camptothecin [123]. SPOP depletion can lead to impaired DDR, resulting in hypersensitivity to IR [123]. However, it is unclear how SPOP terminates DDR upon IR exposure. Furthermore, Boysen et al. demonstrated the role of SPOP in DDR, with SPOP mutations causing homology-directed repair (HDR) of DSB defects by BRCA1 inactivation and RAD51 foci decrease [55]. The researchers found that the loss function of SPOP results in an increased sensitivity to PARP inhibition after DSB induction [55]. Additionally, SPOP can induce non-degradable ubiquitination of HIPK2 and 53BP1, which are both involved in DDR, and degradable ubiquitination of STED2, which regulates the trimethylation of histone H3K36 [64,65,117,118] that is associated with HDR [117,118]. These results suggest that SPOP plays a role in genome stability via multiple pathways affecting DDR.

Gene rearrangements caused by DNA replication stress can also result in genome instability. Previous studies have reported that DSBs induced by the collaboration of TOP2A, 2B and AR signaling lead to gene rearrangements that contribute to genome instability [124,125]. SPOP can remove TOP2A from the TOP2A-DNA cleavage complex in normally growing cells [42]. However, *SPOP* mutations fail to remove TOP2A from DNA, thus causing genome instability.

## 5. The Regulation of SPOP

The regulation of *SPOP expression* at different levels includes DNA methylation that affects transcription, miRNAs that affect translation, and phosphorylation and self-ubiquitination that affect posttranscriptional modifications. 

Hypermethylation is a common modification of DNA that silences the expression of tumor suppressor genes in most cancer types [126]. DNA methylation can regulate tumor suppressor *SPOP* gene transcription by affecting the binding affinity between RXRA and the *SPOP* promoter in CRC and RXRA as a transcription factor [99]. A similar study in LC suggests that the transcription factor C/EBPα can bind the *SPOP* promoter [127]. In non-small cell lung cancer (NSCLC), the combination of this transcriptional regulatory element and DNA methylation regulates the expression and function of *SPOP* [127]. 

miRNAs are small single-stranded non-coding RNAs that have been shown to negatively modulate SPOP expression by targeting the 3′-untranslated region (3′-UTR) of its mRNA. Huang et al. in 2014 showed the post-transcriptional regulation of SPOP expression by miR-145 [128]. In 2018 and 2019, several miRNAs were identified that can regulate SPOP expression in RCC, oral squamous cell carcinoma (OSCC), CRC, NSCLC and GC. In RCC, carcinogenesis is controlled through the E2F1-miR-520/372/373-SPOP axis. The miR-520/372/373 family is downregulated, and SPOP was upregulated, resulting in a significant decrease of PTEN and DUSP7 levels with subsequent increased proliferation, invasion/migration and metastasis [129]. Recently, a study suggested that miRNA-17-5p can target SPOP and that upregulation of miRNA-17-5p can downregulate SPOP to promote proliferation and inhibit anti-tumor immunity in CRC through the overexpression of PD-L1 [130]. Intriguingly, in OSCC, miR373 was upregulated in samples as well as cell lines and negatively regulates the expression of the SPOP protein but not mRNA, promoting proliferation, invasion and migration of corresponding cells [131]. Another study showed that miR-372/373 enhances CRC cell stemness by targeting factors important for cell differentiation, including SPOP [132]. Additionally, miR-543 and miR-520b are upregulated in GC and LC, respectively [133,134]. SPOP has been identified as a direct target of these miRNAs that act to promote proliferation and metastasis by decreasing SPOP expression [133,134]. 

Post-transcriptional modifications of the SPOP protein consist of phosphorylation and self-ubiquitination. Zhang et al. reported that cyclin D-CDK4 can mediate the phosphorylation of SPOP to promote SPOP degradation by regulating PD-L1 protein abundance [66]. More recently, everal have further indicated that SPOP acts as a tumor suppressorhas been shown to stabilize its interaction with 53BP1 [65]. Subsequently, SPOP induces polyubiquitination of 53BP1 and dissociation of 53BP1 from DSBs to activate HDR and consequently induce genome stability [65]. Interestingly, another study found that snails can facilitate SPOP self-ubiquitination and degradation in a cullin3-dependent manner [135]. 

SPOP function can also be regulated by the liquid-liquid phase separation (LLPS) of proteins as a novel regulatory mechanism [136,137]. Increasing evidence has indicated that various nuclear bodies, such as nuclear speckles, DNA damage loci and other substrate-containing bodies, contain the SPOP protein [15,49]. LLPS is critical for SPOP-substrate ubiquitination and the subsequent degradation in cells [136]. In contrast, tumor-associated *SPOP* mutations may disrupt LLPS and thereby inhibit the ubiquitin-dependent proteolysis of substrates.

Mukhopadhyay et al. showed that the GTPase Activating Protein (SH3 Domain) Binding Protein 1 (G3BP1) can interact with SPOP and inhibit SPOP ubiquitin ligase in PrCa [138]. Ji et al. have also found that heparan sulfate 3-O-sulfotransferase 1 (HS3ST1) inhibits SPOP expression and also inhibits activation of the NF-κB pathway activation mediated by FADD degradation in NSCLC [139].

## 6. Summary and Perspectives 

In this review, we summarize the molecular functions and dual roles of SPOP in promoting or inhibiting tumorigenesis in a cancer type-specific manner by targeting multiple proteins. These roles suggest that SPOP and its substrates are potentially viable targets for the development of cancer therapeutics. The complex oncogenic and tumor suppressive roles of SPOP indicate that the design of therapeutic strategies is cancer type specific. For example, SPOP inhibitors could be used to treat kidney cancer as SPOP is an oncogene in RCC. As a putative tumor suppressor in other cancer types, SPOP promoters may also be a desirable approach. An alternative approach is to target the factors upstream of SPOP; for instance, the upregulation of miR-520/372/373 could cause the downregulation of SPOP in CRC cells, inhibiting proliferation, invasion and metastasis by increasing the levels of PTEN and DUSP7 [129]. Conversely, the downregulation of miR-17-5p, miR-543 and miR-520b can inhibit proliferation and metastasis via the upregulation of SPOP [130,133,134]. 

SPOP binds to the cullin 3–RING box 1 through the BTB domain to increase E3 ubiquitin ligase activity by promoting the dimerization of SPOP [40,48]. It would be an attractive option to design small-molecule inhibitors that target cullin 3–RING box 1 to affect its affinity for SPOP and thereby decrease SPOP dimerization. 

The majority of investigations have focused on the role of SPOP in PrCa. Further studies are needed in other cancer types, including the systematic exploration of SPOP substrates. Future studies will undoubtedly provide further insight into the functional mechanisms of SPOP in human cancers and uncover a rationale for the design of therapeutic interventions. 

## Figures and Tables

**Figure 1 genes-13-02051-f001:**
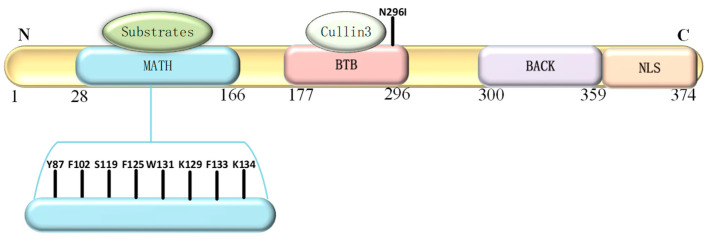
Illustration of the structural features of Speckle-type POZ (pox virus and zinc finger protein) protein (SPOP). The modular structural arrangement of SPOP is shown, which includes the N-terminal meprin and TRAF homology (MATH) domain that selectively recognizes and recruits substrates. Somatic mutations are predominantly clustered in several key amino acids in the N-terminal MATH domain, including Y87, F102, S119, F125, K129, W131, F133 and K134. The Bric-a-brac-Tramtrack/Broad (BTB) domain is responsible for binding to cullin3 and SPOP dimerization, which also involves the BACK domain. The nuclear localization signal (NLS) is a C-terminal nuclear localization sequence.

**Figure 2 genes-13-02051-f002:**
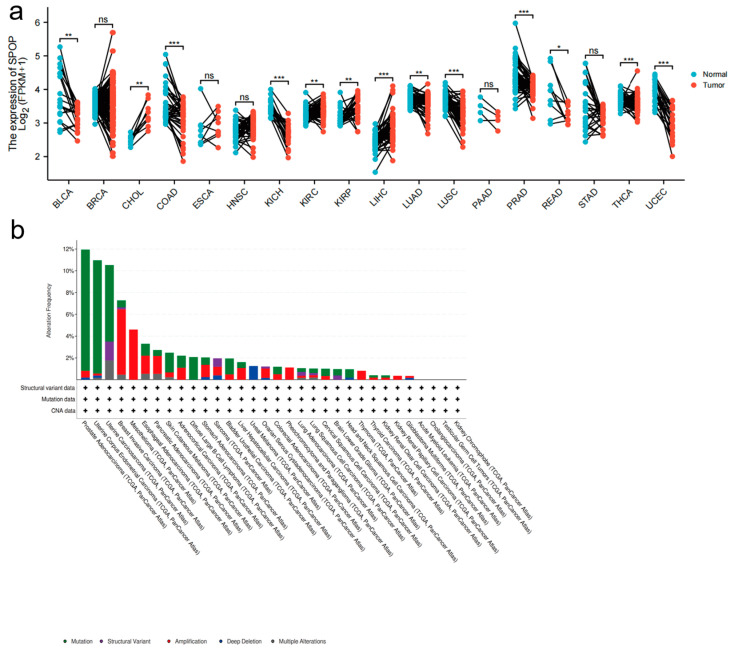
*SPOP* mRNA expression in different cancers. (**a**) A graph showing the levels of *SPOP* mRNA expression levels in a variety of cancers along with corresponding normal controls; BLCA, Bladder Urothelial Carcinoma; BRCA, Breast invasive carcinoma; CHOL, Cholangiocarcinoma; COAD, Colon Adenocarcinoma; ESCA, esophageal carcinoma; HNSC, Head and Neck squamous cell carcinoma; KICH, Kidney Chromophobe; KIRC, Kidney renal clear cell carcinoma; KIRP, Kidney renal papillary cell carcinoma; LIHC, Liver hepatocellular carcinoma; LUAD, Lung adenocarcinoma; LUSC, Lung squamous cell carcinoma; PAAD, Pancreatic adenocarcinoma; PRAD, Prostate adenocarcinoma; READ, Rectum adenocarcinoma; STAD, Stomach adenocarcinoma; THCA, Thyroid carcinoma; UCEC, Uterine Corpus Endometrial Carcinoma; ns, not significant; * *p* < 0.05, ** *p* < 0.01, *** *p* < 0.001. (**b**) *SPOP* alterations occur across multiple cancer types (data from the cBioPortal database). (**c**) The correlation of SPOP with overall survival is further elevated (data from the The Cancer Genome Atlas (TCGA) portal or other published datasets of clinical cancer samples as indicated).

**Figure 3 genes-13-02051-f003:**
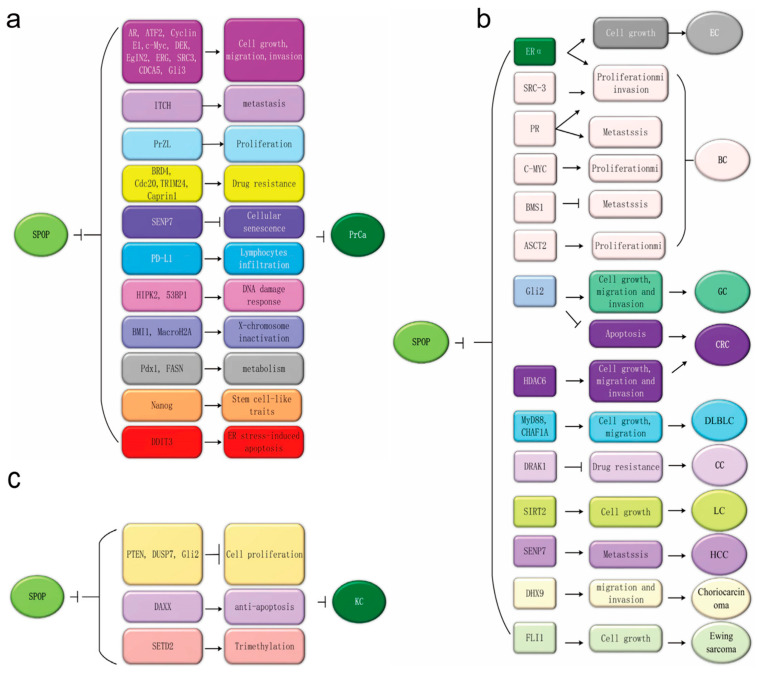
The potential oncogenic functions of Speckle-type POZ (pox virus and zinc finger protein) protein (SPOP) in kidney cancer and tumor-suppressive roles in other cancer types. (**a**) SPOP exerts potential antitumor activity by regulating proliferation, migration, invasion, metastasis, apoptosis, drug resistance, cell senescence and lymphocyte infiltration via modulating various substrates in PrCa. (**b**) SPOP has tumor-suppressive effects across multiple cancer types, including endometrial, gastric, colorectal, cervical and lung cancers, hepatocellular carcinoma, choriocarcinoma, Ewing sarcoma and diffuse large B-cell lymphoma. It has dual roles in breast cancer that regulate the degradation of its various substrates. (**c**) SPOP exerts its potential oncogenic activity by enhancing the ubiquitination and degradation of Phosphatase and tensin homolog (PTEN), Dual-Specificity Phosphatase 7 (DUSP7), Gli2 and SETD2 to promote renal carcinogenesis.

**Table 1 genes-13-02051-t001:** Summary of SPOP substrates and their cellular functions. Androgen receptor—AR, progesterone receptor—PR, steroid receptor coactivator 3—SRC3, Macrohistone H2A1—MacroH2A, Phosphatase and tensin homolog—PTEN, Dual-Specificity Phosphatase 7—DUSP7, Fas-associated death domain protein—FADD, Programmed death ligand 1—PD-L1, Pancreatic duodenal homeobox 1—Pdx1, fatty acid synthase—FASN, breast cancer metastasis suppressor 1—BRMS1.

SPOP-Binding Substrates	Pathway or Process Involved	Refs.
AR, TRIM24	Androgen receptor signaling	[56,57,59]
17βHSD4	Androgen biosynthesis	[88]
ERα	Estrogen receptor signaling	[35,95,96]
DDIT3	ER stress-induced apoptosis	[78]
PR	Progesterone receptor signaling	[25,36,37]
SRC3	Progesterone, estrogen and androgen receptor signaling	[25,58]
HIPK2, 53BP1	DNA damage response	[64,65]
GLP	DNA methylation	[90]
SETD2	H3K36 Trimethylation	[117,118]
DEK	DNA replication and mRNA processing	[77]
BMI1, MacroH2A	X-chromosome inactivation	[67,68]
Gli2, Gli3	Hedgehog signaling pathway	[39,60,61,62,63,98,99]
CDCA5, PDK1, PTEN	PI3K-AKT signaling pathway	[39,91,92]
DUSP7	ERK pathway	[39]
PIPKIIβ	Phosphoinositide signaling pathway	[113]
MyD88	TLR-MyD88-NF-κB pathway	[101,102,103]
FADD	Pancreatic stellate cell activation and NF-κβ signaling	[105]
PD-L1	PD-L1/PD-1 pathway	[66]
BRD2/3/4	AKT-mTORC1 signaling pathway	[19,75,76]
Cdc20, CYCLIN E1	Cell cycle	[81,82]
c-MYC	Cell cycle, apoptosis and cellular transformation	[36,83]
Daxx	Apoptosis, transcriptional regulation, cell cycle and angiogenesis	[39]
SENP 7	Cellular senescence	[72,107]
HDAC6, ITCH, DHX9	Epithelial-mesenchymal transition	[89,100,108]
Pdx1	Glucose homeostasis and pancreatic β-cell development	[69,70,71]
Nanog	Stem cell-like traits	[79,80]
INF2	Mitochondrial fission	[84]
FASN	lipid metabolism	[87]
Caprin1	Stress granule (SG) assembly	[52]
BRMS1	Breast cancer metastasis	[38]
SLC7A1	Arginine metabolism	[110]

## Data Availability

Data is contained within the article.

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
