# Peer review of "SPOP in Cancer: Phenomena, Mechanisms and Its Role in Therapeutic Implications"

_genes, 2022, doi:10.3390/genes13112051_

Round 1
Reviewer 1 Report
In the manuscript “SPOP in cancer: phenomena, mechanisms and its role in therapeutic implications”, Yang et al systemically summarize the basic knowledge of SPOP and its role in cancer. The topic is interesting, but it is not well-organized and written. It’s difficult to follow and understand what the author wants to express. In addition, there has been a similar review paper “SPOP and cancer: a systematic review” published in 2020. What’s new compared with this one? There are several suggestions to help the authors improve their manuscript.
1. The writing of this manuscript should be improved remarkably.
2. The font size of all figures should be consistent, at least similar. The authors should modify their figures and let them neatly and attractively with professional software, such as Adobe Illustrator. Clearly defined figures with artistic sense will improve the readability of the manuscript.
3. In Figure 2A, the means of the symbol star and “ns” should be provided in the figure legends. The full names of the tumor/cancer are also required.
4. In Figure 2C, how to definite the cut-off value? It seems the authors chose different cut-off values among those different types of cancer. In general, the median is the best one in the survival analysis.
Reviewer 2 Report
In this manuscripts entitled ‘SPOP in cancer: phenomena, mechanisms and its role in therapeutic implications’, the authors summarised about the function of SPOP and analysed some data by using public database such as TCGA.This review is well summarised and will be a good map for this field.
It would be more polished if they approach some points mentioned below.
In figure 1, a tiny ‘N’ is posted below BTB domain. What this ‘N’ means?
In figure 2a, they showed the expression change between normal tissue and tumour by using cBioportal database. 13 tumours have statistically significant differences against their normal tissues. It would be informative that each prognosis is analysed by TCGA with correspondence to these 13 tumours. Prognosis of only four types of cancer were showed in figure 2c.
All acronyms in figure 2a should be defined elsewhere in this manuscript.
In line 368, gramatical editing is recommended.
